

# Hematological convergence between Mesozoic marine reptiles (Sauropterygia) and extant aquatic amniotes elucidates diving adaptations in plesiosaurs

Corinna V. Fleischle[1,*], P. Martin Sander[1,2], Tanja Wintrich[1,3] and Kai R. Caspar[1,4,*]

[1] Section Paleontology, Institute of Geosciences, University of Bonn, Bonn, Germany
[2] Dinosaur Institute, Natural History Museum of Los Angeles County, Los Angeles, CA, USA
[3] Institute of Anatomy, University of Bonn, Bonn, Germany
[4] Department of General Zoology, Faculty of Biology, University of Duisburg-Essen, Essen, Germany
* These authors contributed equally to this work.

Corresponding authors
Corinna V. Fleischle,
corinna@fleischle.de
Kai R. Caspar,
kai.caspar@uni-due.de

## ABSTRACT

Plesiosaurs are a prominent group of Mesozoic marine reptiles, belonging to the more inclusive clades Pistosauroidea and Sauropterygia. In the Middle Triassic, the early pistosauroid ancestors of plesiosaurs left their ancestral coastal habitats and increasingly adapted to a life in the open ocean. This ecological shift was accompanied by profound changes in locomotion, sensory ecology and metabolism. However, investigations of physiological adaptations on the cellular level related to the pelagic lifestyle are lacking so far. Using vascular canal diameter, derived from osteohistological thin-sections, we show that inferred red blood cell size significantly increases in pistosauroids compared to more basal sauropterygians. This change appears to have occurred in conjunction with the dispersal to open marine environments, with cell size remaining consistently large in plesiosaurs. Enlarged red blood cells likely represent an adaptation of plesiosaurs repeated deep dives in the pelagic habitat and mirror conditions found in extant marine mammals and birds. Our results emphasize physiological aspects of adaptive convergence among fossil and extant marine amniotes and add to our current understanding of plesiosaur evolution.

## INTRODUCTION

The Sauropterygia arguably were the most successful clade of marine reptiles in the Mesozoic Era (*Motani, 2009*; *Kelley & Pyenson, 2015*; *Renesto & Dalla Vecchia, 2018*). The most speciose and long-lived taxon among sauropterygians were the Eosauropterygia, which emerged in the Early Triassic (*Rieppel, 2000*; *Benson, Evans & Druckenmiller, 2012*; *Jiang et al., 2014*; *Renesto & Dalla Vecchia, 2018*). This clade traditionally includes two major groups, the small-bodied Pachypleurosauridae, whose monophyly is debated (*Holmes, Cheng & Wu, 2008*; *Klein, 2010*), and the larger, morphologically more diverse
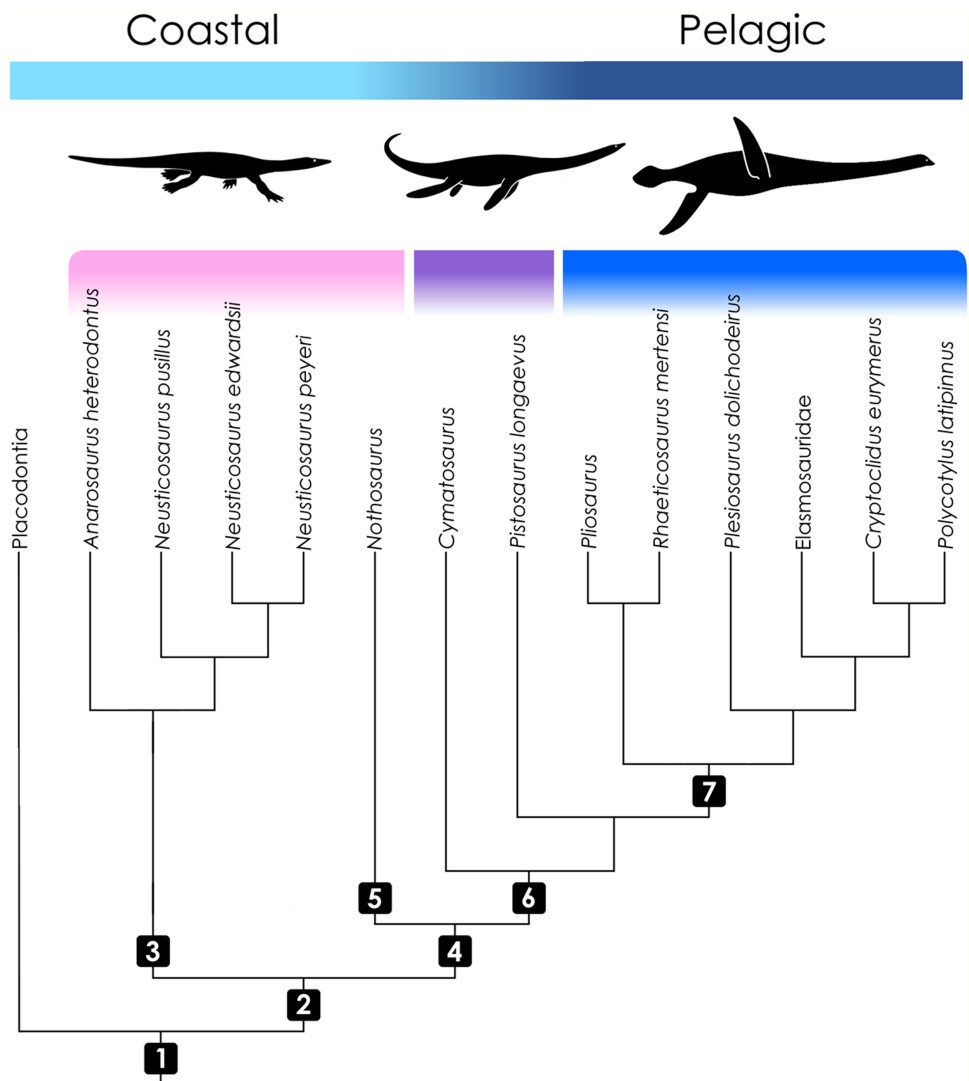

**Figure 1 Cladogram of taxa included in the study with information on ecology.** Topology follows *Rieppel (2000)*, *Ketchum & Benson (2010)*, and *Wintrich et al. (2017)*. Color coding indicates the operational groups considered herein. The pink bar denotes basal eosauropterygian groups (Pachypleurosauridae, Nothosauroidea) from coastal and shallow-water habitats. The violet bar marks the paraphyletic Pistosauridae in which notable adaptations to offshore environments were acquired. The blue bar denotes the derived pelagic taxon Plesiosauria. Numbers indicate inclusive taxa: (1) Sauropterygia; (2) Eosauropterygia; (3) Pachypleurosauridae; (4) Eusauropterygia; (5) Nothosauroidea; (6) Pistosauroidea; (7) Plesiosauria. Silhouettes by Kai R. Caspar.

Eusauropterygia (*Rieppel, 2000*) (Fig. 1). These, in turn, comprise the Nothosauroidea and Pistosauroidea. While the former went extinct before the end of the Triassic, the Pistosauroidea persisted until the K-Pg boundary (*Rieppel, 2000*; *Ketchum & Benson, 2010*; *Benson, Evans & Druckenmiller, 2012*; *Benson & Druckenmiller, 2014*). Pistosauroids are most prominently represented by the iconic Plesiosauria, the only sauropterygian group that survived the Triassic–Jurassic mass extinction (*Benson, Evans & Druckenmiller, 2012*; *Wintrich et al., 2017*) and continued to be highly successful throughout the

Mesozoic. Apart from the plesiosaur radiation, pistosauroids include the basal paraphyletic non-plesiosaurian forms, herein referred to as Pistosauridae.

While pachypleurosaurs and nothosauroids inhabited shallow coastal waters, the more derived pistosauroids were largely pelagic animals, populating predominately offshore habitats (*Sues, 1987*; *Krahl, Klein & Sander, 2013*). This transition from coastal to pelagic ecosystems is widely acknowledged as an important event in sauropterygian evolution (*Benson, Evans & Druckenmiller, 2012*). Most notably, the shift in habitat preference is coupled with the emergence of the characteristic plesiosaurian bauplan which is foreshadowed in the Pistosauridae (*Sues, 1987*). It is characterized by a complete transformation of the extremities to stiff flippers, a shortening of the trunk and tail, and an elongation of the neck (the latter is secondarily shortened, however, in several derived plesiosaurian groups). These characters correspond to a unique mode of paraxial locomotion ("four-winged under-water flight"), which enabled plesiosaurs to effectively propel themselves through the water combined with great maneuverability. Shallow-water eosauropterygians, on the other hand, swam by axial undulation supported by the limbs to varying degrees (e.g., Nothosauroidea, basal Pistosauroidea) (*Zhang et al., 2014*; *Klein et al., 2015*).

The physiological consequences of offshore environment colonization in sauropterygians remain largely unexplored. Basal eosauropterygians, such as pachypleurosaurids, already reproduced and presumably spent their whole life in coastal waters (*Sander, 1989*; *Cheng, Wu & Ji, 2004*). Still, the available data suggest important physiological changes in response to the adaptation to open marine habitats in more derived groups. Qualitative and quantitative osteohistological investigations of eosauropterygians inferred elevated metabolic rates for pistosauroids, thereby suggesting endothermy in this group (*Klein, 2010*; *Krahl, Klein & Sander, 2013*; *Wintrich et al., 2017*; *Fleischle, Wintrich & Sander, 2018*). These results conform to those from studies on isotope composition of plesiosaurian tooth phosphate (*Bernard et al., 2010*). Enhanced metabolic rates apparently facilitated dispersal into pelagic habitats around the globe (*Krahl, Klein & Sander, 2013*; *Wintrich et al., 2017*) and evolved convergently in other marine reptile groups as well (*Bernard et al., 2010*). Apart from that, the morphology of the endosseous labyrinth in diverse sauropterygians traces the shift in locomotory style subsequent to the colonization of marine habitats (*Neenan et al., 2017*). Pistosauroids gradually evolved a distinct compact inner ear morphology similar to extant marine turtles, while the inner ear of basal sauropterygians more closely resembles the condition found in extant crocodiles and marine iguanas (*Neenan et al., 2017*). This indicates a more sophisticated diving profile in the former group. In accordance with this, avascular necrosis has repeatedly been reported for pistosauroids throughout their evolutionary history (*Rothschild & Storrs, 2003*; *Surmik et al., 2017*). In extant tetrapods, this type of bone tissue lesion is indicative of decompression syndrome (*Carlsen, 2017*). In pachypleurosaurs and nothosauroids, these lesions are almost completely absent (*Rothschild & Storrs, 2003*), again suggesting contrasting lifestyles and diving behavior in these basal groups compared to pistosauroids.

So far, physiological adaptations to pelagic lifestyles on the cellular level received no attention in sauropterygians or other fossil marine reptiles. Obviously, data on
cellular characteristics in most fossil vertebrates have to be inferred from bone tissue. In petrographic thin-sections of fossil bone, its microstructure, including vascularization, can be studied in detail. The caliber of the smallest vascular canals found in bone tissue tightly correlates with the size of the erythrocytes, the oxygen-transporting red blood cells (RBC) of the respective species, allowing the reconstruction of RBC size in extinct taxa via regression models (*Huttenlocker & Farmer, 2017*). Due to their pivotal role in systemic oxygen transport, RBC size can potentially provide further information on pelagic adaptations in sauropterygians.

In the context of marine mammal research, it has been hypothesized that secondarily aquatic species tend to evolve enlarged RBCs (*Wickham et al., 1989*; *Ridgway et al., 1970*). Large RBCs are expected to store increased amounts of hemoglobin to allow for persistent tissue oxygen supply during prolonged dives, providing adaptive advantages for pelagic species (*Wickham et al., 1989*; *Promislow, 1991*). However, comparisons between RBC parameters in marine amniotes and their terrestrial relatives have only superficially been undertaken, and potential patterns of convergence remain unappreciated (compare *Hawkey, 1975*). In general, mammals have the smallest RBCs among amniotes related to the evolutionary loss of the nucleus and avian RBC size is lower than in modern reptiles, presumably because of the general inverse relationship between metabolic rate and RBC size observed in vertebrates (*Gregory, 2002*).

In this study, we analyze osteohistological features of diverse eosauropterygian taxa to trace RBC size evolution across the nothosaurian–pistosaurian transition in order to identify correlates of pelagic adaptation. We hypothesized that pistosauroids have relatively larger vascular canals indicative of enlarged RBCs compared to pachypleurosaurs and nothosauroids. This condition would be in accordance with advanced pelagic adaptations in the former group. To track RBC size evolution, we apply phylogenetic eigenvector maps (PEM) (*Guénard, Legendre & Peres-Neto, 2013*). This technique is increasingly used in recent histomorphometric studies (*Legendre et al., 2016*; *Olivier et al., 2017*; *Fleischle, Wintrich & Sander, 2018*) and allows estimating unknown trait values from a predictor variable while taking into account phylogenetic relationships. Inferring hematological parameters to deduce ecophysiological adaptations is a novel approach which has not been considered in marine reptile paleobiology before. To complement our inferred data on sauropterygians and to test for influences of body size and ecology on RBC size parameters in extant groups, we additionally compiled RBC size measurements for modern reptiles, birds and mammals with a focus on marine groups.

Different RBC size proxies pertain in the literature, at times hindering effective comparisons. Whereas cell volume would appear to be the most useful proxy and indeed has been widely used (see below), other size proxies are two-dimensional ("area") or one-dimensional ("width", "length"). We employed all of these size proxies in our study, the choice depending on availability of comparative data sets. Vertebrate RBC shape is typically that of an oblate to scalenoid spheroid (*Gulliver, 1862*). The most common proxies used to describe RBC size are "area", "width" and "length". "Area" describes the lateral surface area of the disc-shaped erythrocyte as seen under a light microscope. "Length" corresponds to the longest axis (diameter) that can be drawn on the RBC,

**Table 1** List of eosauropterygian specimens studied.

| Species | Higher taxon | Specimen number | Skeletal element | Geological time | Previously studied by |
|---|---|---|---|---|---|
| *Anarosaurus heterodontus* | Pachypleurosauridae | NMNHL Wijk. 06-38fe | Femur | Middle Triassic | *Klein (2012)*, *Fleischle, Wintrich & Sander (2018)* |
| *Neusticosaurus edwardsii* | Pachypleurosauridae | PIMUZ T3455 | Humerus | Middle Triassic | *Sander (1989, 1990)*, *Fleischle, Wintrich & Sander (2018)* |
| *Neusticosaurus peyeri* | Pachypleurosauridae | PIMUZ T 4089 | Humerus | Middle Triassic | *Sander (1989, 1990)* |
| *Neusticosaurus pusillus* | Pachypleurosauridae | PIMUZ T 3566 | Humerus | Middle Triassic | *Sander (1989, 1990)* |
| *Nothosaurus* sp. | Nothosauroidea | IGWH 21 | Femur | Middle Triassic | *Klein (2010)*, *Fleischle, Wintrich & Sander (2018)* |
| *Cymatosaurus* sp. | Pistosauroidea indet. (pistosaurid grade) | IGWH 6 | Humerus | Middle Triassic | *Klein (2010)* |
| *Pistosaurus longaevus* | Pistosauridae | SMNS 84825 | Humerus | Middle Triassic | *Krahl, Klein & Sander (2013)*, *Fleischle, Wintrich & Sander (2018)* |
| *Cryptoclidus eurymerus* | Plesiosauria: Cryptoclididae | IGPB R 324 | Femur | Middle Jurassic | *Wintrich et al. (2017)*, *Fleischle, Wintrich & Sander (2018)* |
| Elasmosauridae indet. | Plesiosauria: Elasmosauridae | OMNH MV 85 | Humerus | Late Cretaceous | *Wintrich et al. (2017)*, *Fleischle, Wintrich & Sander (2018)* |
| *Plesiosaurus dolichodeirus* | Plesiosauria: Plesiosauridae | IGPB R90 | Femur | Early Jurassic | *Wintrich et al. (2017)*, *Fleischle, Wintrich & Sander (2018)* |
| *Pliosaurus* sp. | Plesiosauria: Pliosauridae | SMNS 96896 | Femur | Middle Jurassic | |
| *Polycotylus latipinnus* | Plesiosauria: Polycotylidae | LACM 129639A ("Mom") | Femur | Late Cretaceous | *O'Keefe et al. (2019)* |
| *Rhaeticosaurus mertensi* | Plesiosauria: Pliosauridae | LWL-MfN P 64047 section PM3 | Femur | Late Triassic | *Wintrich et al. (2017)*, *Fleischle, Wintrich & Sander (2018)* |

Note:
Collection Acronyms: IGWH, Institut für Geowissenschaften, University of Halle-Wittenberg, Halle, Germany; LWL-MFN, LWL-Museum für Naturkunde, Münster, Germany; NMNHL, National Museum of Natural History (NCB Naturalis), Leiden, The Netherlands; OMNH, Osaka Museum of Natural History, Osaka, Japan; PIMUZ, Paläontologisches Institut und Museum Universität Zürich, Zurich, Switzerland; SMNS, Staatliches Museum für Naturkunde, Stuttgart, Germany; IGPB, Steinmann Institute Paleontology Collection, University of Bonn, Bonn, Germany; LACM, Natural History Museum of Los Angeles County, Los Angeles, USA.

while "width" denotes the shortest one (*Hartman & Lessler, 1963*). RBC volume is either measured directly or can be calculated based on the other proxies, as done in the current study for fossil species.

# MATERIALS AND METHODS

## Fossil sample base

We studied petrographic histological thin-sections of fossil bones from 13 eosauropterygian taxa, most of which were already included in earlier studies (Table 1). Among the basal Eosauropterygia, we examined several species of pachypleurosaurids

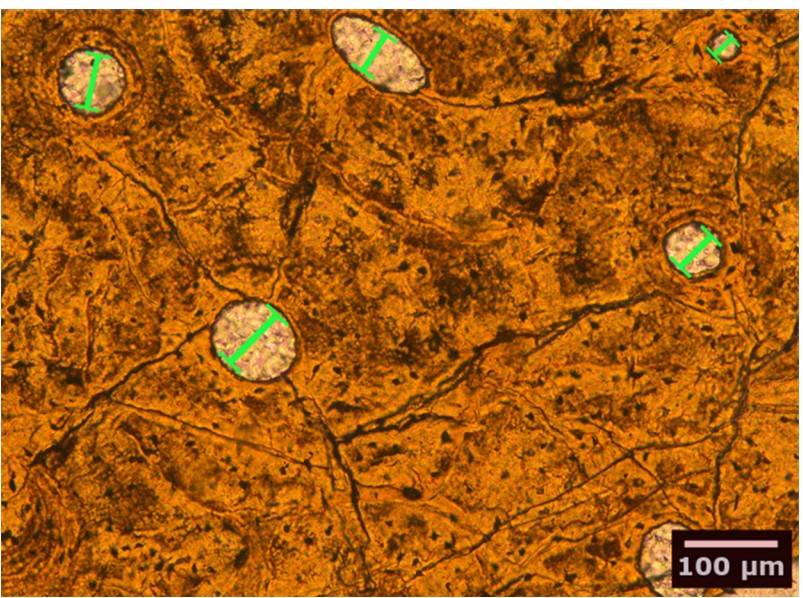

**Figure 2 Bone histological thin-section of the femur of the plesiosaur *Pliosaurus* sp.** The width (smallest diameter, green bars) of longitudinal vascular canals and nodes in reticular canals found in the bone matrix were measured.

(*Anarosaurus heterodontus*, *Neusticosaurus edwardsii*, *Neusticosaurus peyeri*, and *Neusticosaurus pusillus*) and a nothosaurid (*Nothosaurus* sp.). Among Pistosauroidea, we included the basal taxa *Cymatosaurus* sp. and *Pistosaurus longaevus* ("Pistosauridae") as well as diverse plesiosaurs (*Plesiosaurus dolichodeirus*, *Rhaeticosaurus mertensi*, *Pliosaurus* sp., *Cryptoclidus eurymerus*, *Polycotylus latipinnus*, and an indeterminate Japanese elasmosaurid). We collected histomorphometric data in petrographic thin sections of 50–80 μm thickness from stylopodial (humerus/femur) mid-diaphyses by analyzing microscopic images taken with a Leica DFC420 color camera mounted on a polarizing microscope (Leica DM2500LP) using the software EASYLAB 7 (Fig. 2). We also took overview images of thin-sections with an Epson V750 scanner. Contrasting with stylopodials from other marine amniotes, such as cetaceans and ichthyosaurs, the respective bones in plesiosaurs do not show an increase in the amount of primary cancellous bone at the expense of a compact cortex.

## Measurement and inference of eosauropterygian RBC size

In their study, *Huttenlocker & Farmer (2017)* found a correlation of minimum and mean vascular canal dimensions with RBC size (area and width) in extant amniotes. Applying the R package MPSEM (*Guénard, Legendre & Peres-Neto, 2013*), we converted the phylogeny of the extant species (adopted from *Huttenlocker & Farmer, 2017*) into PEMs to build the predictive models. Both potential predictor variables, that is, minimum and mean canal caliber, and the estimated variables (RBC area and width) (all data taken from *Huttenlocker & Farmer, 2017*) were log-transformed to adjust for the large range of values in the data set. We selected minimum canal caliber as the best predictor variable

based on the Akaike information criterion, corrected for small sample size (AICc) (*Burnham, Anderson & Huyvaert, 2011*) and cross-validated using leave-one-out cross-validation, suitable for a small training data set.

The fossil eosauropterygians were then added to the tree of predicting species. Since a sister group relationship of Sauropterygia and Lepidosauromorpha is commonly accepted (*Rieppel, 2000*; *Chen et al., 2014*; but see *Kelley & Pyenson, 2015*), it was adopted in our study. For internal eosauropterygian relationships, we used the phylogeny of *Rieppel (2000)*, depicting a monophyletic Pachypleurosauridae as the sister group to Eusauropterygia which includes *Nothosaurus* and Pistosauroidea (Fig. 1). For pistosauroid ingroup relationships, were entered a topology based on *Ketchum & Benson (2010)* and *Wintrich et al. (2017)*. Using the model, we estimated RBC area and width for each fossil specimen (Table 1), including the 95% confidence intervals. For the statistical comparison of basal sauropterygians and pistosauroids, we used a Welch two sample *t*-test.

For the comparison of the fossils with the RBC volume data sets for extant taxa compiled by us, we calculated RBC volume ($V$), from estimated RBC width and area ($A$) of the fossils. We approximated sauropterygian RBC shape as a scalenoid spheroid, that is, a spheroid that has three different axes, because this is the shape of modern RBCs. The major axis is length ($a$), the intermediate axis is width ($b$), and the minor axis is ($c$).

In a first step, we calculated $a$ as

$$a = (4\,A)/(b\,\pi)$$

For calculating $V$, we made one additional assumption, that is, that $c$ is half of $b$, that is, that the minor axis is half the length of the intermediate axis. Using length $a$, width $b$, and minor diameter $c$, we calculated volume $V$ as follows

$$V = (1/6)\,\pi\,a\,b\,c$$

Modelling and calculations were performed in R (*R Core Team, 2017*).

## RBC and body mass parameters in extant taxa

To analyze potential influences of body mass and ecology on RBC size in different clades and between taxonomic ranks, we compiled a dataset of RBC size parameters (area, width, length, depending on the available data) and body mass for 188 species of extant reptiles (lepidosaurs, turtles, and crocodiles) from the literature (see Tables S6 and S7). Given their phylogenetic affiliation, such patterns in extant reptiles, especially in lepidosaurs, can be hypothesized to bear relevant implications for sauropterygians. In addition to this, we collected published RBC volume data on selected marine mammals ($n = 28$) and birds ($n = 6$) as well as non-marine representatives of these groups ($n_{Mammalia} = 82$; $n_{Aves} = 36$) (see Tables S8 and S9) in order to further test if adaptation to pelagic life correlates with specific trends in amniote RBC size evolution. Variance in the compiled data sets was assessed by means of the Kruskal–Wallis test, and differences between marine and non-marine groups were assessed by applying the Welch two-sample *t*-test. In case several measurements were found for the same species, data were averaged.

**Table 2 Estimates of different RBC size proxies in Eosauropterygia.**

| Species | Higher taxon | Area (μm²) | Width (μm) | Length (μm) | Volume (μm³) |
|---|---|---|---|---|---|
| *Anarosaurus heterodontus* | Pachypleurosauridae | 78.8 | 8.0 | 12.54 | 210.1 |
| *Neusticosaurus edwardsii* | Pachypleurosauridae | 82.8 | 8.2 | 12.86 | 220.9 |
| *Neusticosaurus peyeri* | Pachypleurosauridae | 72.4 | 7.7 | 11.97 | 186.7 |
| *Neusticosaurus pusillus* | Pachypleurosauridae | 79.3 | 8.0 | 12.62 | 211.5 |
| *Nothosaurus* sp. | Nothosauroidea | 65.75 | 7.3 | 11.47 | 159.4 |
| *Cymatosaurus* sp. | Pistosauridae | 96.5 | 8.8 | 13.97 | 284.4 |
| *Pistosaurus longaevus* | Pistosauridae | 122.7 | 10.0 | 15.62 | 405.7 |
| Elasmosauridae indet. | Plesiosauria | 156.4 | 11.3 | 17.6 | 585.8 |
| *Cryptoclidus eurymerus* | Plesiosauria | 123.4 | 10.0 | 15.7 | 411.6 |
| *Plesiosaurus dolichodeirus* | Plesiosauria | 159.8 | 11.4 | 17.9 | 602.7 |
| *Pliosaurus* sp. | Plesiosauria | 220.9 | 13.4 | 21.0 | 987.2 |
| *Polycotylus latipinnus* | Plesiosauria | 140.1 | 10.7 | 16.7 | 498.4 |
| *Rhaeticosaurus mertensi* | Plesiosauria | 140.2 | 10.7 | 16.7 | 494.6 |

**Note:**
Area and width were estimated based on the *Huttenlocker & Farmer (2017)* data set for extant species. Volumes and lengths were calculated from area and width.

# RESULTS

## Red blood cell size in Eosauropterygia

Estimated RBC size as expressed by RBC area for the basal sauropterygian pachypleurosaurids and *Nothosaurus* is consistently small (species means ranging from 65.75 to 82.84 μm²; group mean: 75.81 μm²; Table 2; Fig. 3) compared to pistosauroids. For the latter, inferred RBC area is notably larger (species means ranging from 96.45 to 220.93 μm²; group mean: 144.98 μm²; Table 2; Fig. 3). The Welch two sample $t$-test, comparing estimated RBC areas of the basal sauropterygians and pistosauroids, respectively, yields significant differences between the two groups ($t = -5.1768$, $p < 0.001$, df = 8). Concerning group average RBC volumes, an increase of 270% from pachypleurosaurids and *Nothosaurus* (group mean: 197.7 μm³) to the more derived pistosauroids (group mean: 533.8 μm³) was obtained. For measured canal calibers see Table S1, for PEM models and model coefficients for RBC parameter inference, see Tables S2 and S3; for estimated RBC size proxies and confidence intervals, see Tables S4 and S5.

## Body mass and RBC size in extant reptiles

We found evidence for a weak influence of body mass on RBC size in reptiles using the size proxies area and length among extant reptile species (Fig. 4). When data of lepidosaurs, turtles and crocodylians are combined, a weak but statistically highly significant correlation emerges (area: adjusted $R^2 = 0.104$, $p$-value = 0.00017, df=120; length: adjusted $R^2 = 0.3373$, $p$-value <0.00001, df = 178; Fig. 4). At lower taxonomic ranks however, this correlation was not consistently recovered (Fig. S1). For example, turtle RBC length (Testudines, $n = 31$, adjusted $R^2 = 0.488$, $p = 0.02$) and RBC area in true lizards (Lacertidae, $n = 21$, adjusted $R^2 = 0.218$, $p = 0.02$) correlated significantly with body mass, whereas RBC
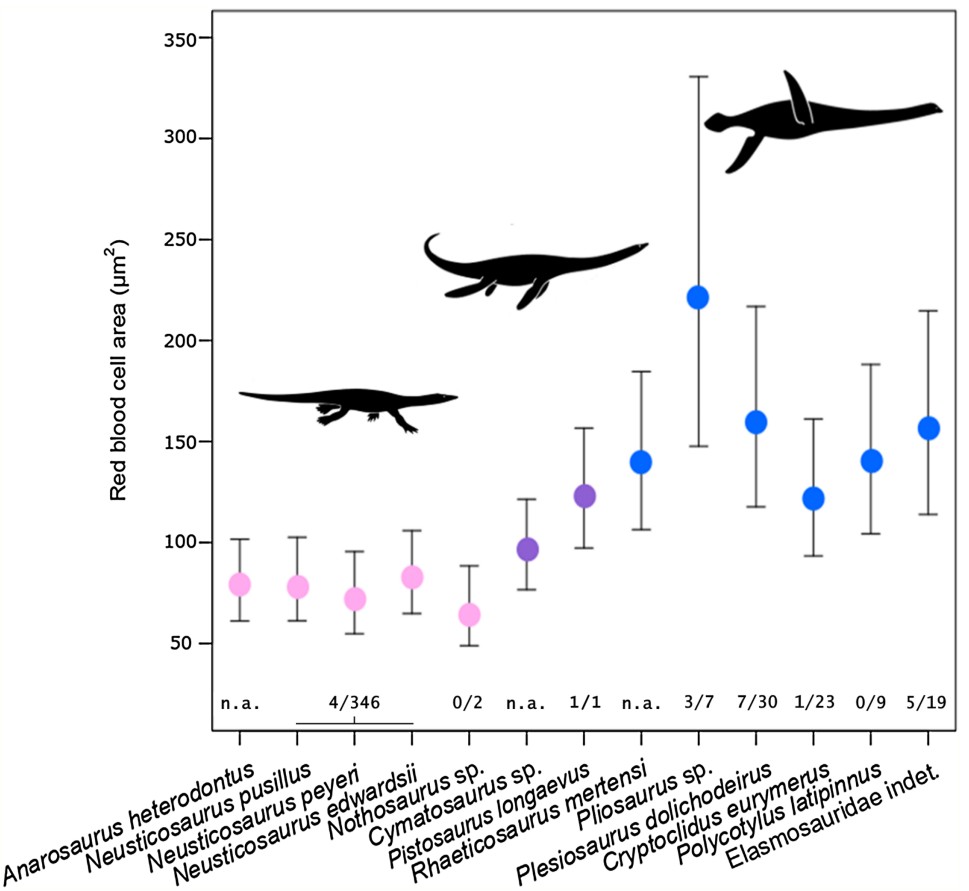

**Figure 3 Estimated RBC area of 13 eosauropterygians, error bars indicating 95% confidence intervals.** Pachypleurosaurids and *Nothosaurus* (pink) have small cells, whereas pistosauroids (Pistosauridae: purple; Plesiosauria: blue) have significantly larger RBCs. Numbers below error bars indicate frequency of propodial head subsidence diagnostic of avascular necrosis in eosauroperygian humeri suggestive of dysbaric stress experienced during deep dives. Data derive from *Rothschild & Storrs (2003)* and *Surmik et al. (2017)* and are presented for the genus level, except for Elasmosauridae, since the sampled specimen is of ambiguous generic identity. Corresponding to the latter, data of all elasmosaurids listed in *Rothschild & Storrs (2003)* are combinedly presented (excluding *Colymbosaurus* and *Muraenosaurus*). Silhouettes by Kai R. Caspar.

area in colubrid snakes instead showed an inverse statistical trend (Colubridae, $n = 42$, adjusted $R^2 = -0.018$, $p = 0.48$) (Fig. S1).

## RBC size and aquatic adaptation in extant taxa

An increase in RBC size in marine taxa compared to related terrestrial groups was consistently found among secondarily aquatic amniotes (Fig. 5). Within this comparison, volume is the RBC size proxy for mammals and birds, and area is the size proxy for reptiles. All aquatic mammal clades exhibit RBC volumes significantly above the terrestrial mammal mean, which was recovered as 64 $\mu m^3$ based on data from a selection of terrestrial mammal species ($n = 82$; SD: 25.19; Table S8). In seals (Pinnipedia; $n = 12$), the mean RBC volume is 127.8 $\mu m^3$ (SD: 26.00), which equals 195.6% of the average volume in closely related non-marine carnivoran species of the superfamily Canoidea, in which

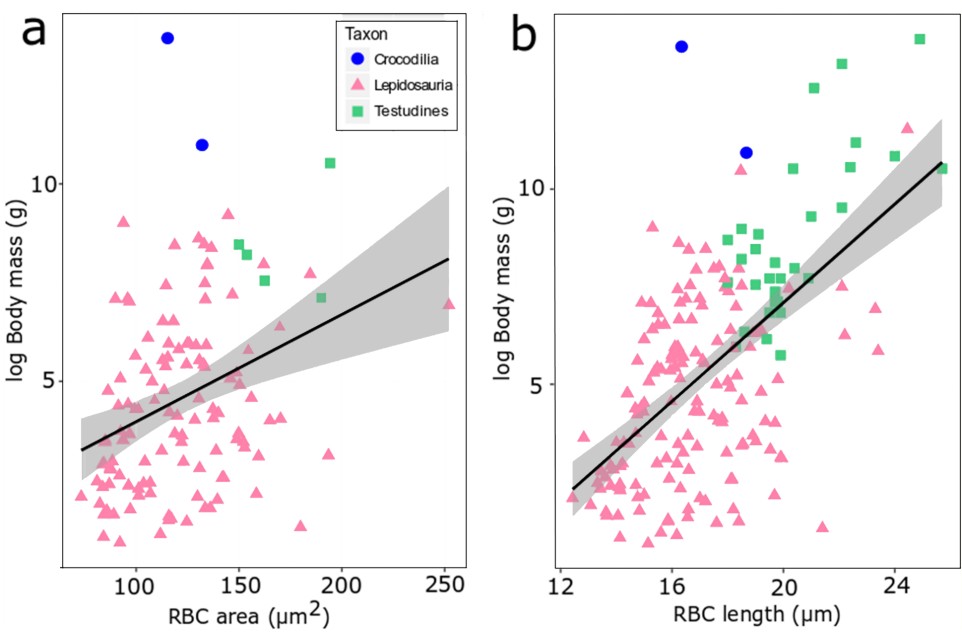

**Figure 4 RBC area (A) and length (B) regressed against log body mass for 188 species of extant reptiles.** Crocodilia are plotted in blue, Lepidosauria in pink, and Testudines in green. The correlation is weak but statistically highly significant (a: adjusted $R^2$ = 0.104, $p$-value = 0.00017, df = 120, b: adjusted $R^2$ = 0.3373, $p$-value < 0.00001, df = 178).

pinnipeds are nested ($n$ = 15; mean: 65.35 $\mu m^3$; SD: 9.33), and 199.7% of the terrestrial mammal mean value, respectively (Fig. 5). The two sampled pinniped families differ notably in RBC size, with members of the Otariidae ($n$ = 4, range: 97–108 $\mu m^3$) displaying smaller RBCs than the ones of the Phocidae ($n$ = 8; range: 105–176 $\mu m^3$). Besides pinnipeds, the RBC volume of the sea otter (*Enhydra lutris*) (113 $\mu m^3$) is also strongly increased (172.9%) compared to non-marine canoids. In cetaceans ($n$ = 14; mean: 121.28 $\mu m^3$; SD: 24.58), the mean RBC volume even was 297% of that of ruminant artiodactyls (Ruminantia; $n$ = 16; mean: 40.83 $\mu m^3$; SD: 15.19), which are among the whales' closest living relatives, and 189.5% that of the terrestrial mammal mean (Fig. 5). RBC volume ranges of terrestrial and aquatic carnivorans as well as whales and terrestrial ungulates do not overlap. While clearly deviating from the ones of their close extant relatives as well as from the mammalian mean ($p$ < 0.001 for all comparisons), RBC volumes in pinnipeds and cetaceans do not significantly differ from each other ($p$ = 0.052). Data on sirenians could only be obtained for one species, *Trichechus manatus*, the mean RBC volume of which is also notably large at 132.6 $\mu m^3$ (*Medway, Black & Rathbun, 1982*) (Fig. 5).

Like marine mammals, penguins, as diving marine birds, exhibit markedly enlarged RBCs compared to other birds (Fig. 5). However, compared to the mammalian groups, the relative increase in RBC volume is less pronounced. The mean RBC volume of penguins (Sphenisciformes; $n$ = 6; mean: 239.7 $\mu m^3$; SD: 26.24) is 140% of that of closely related sea birds (Aequornithes sensu *Burleigh, Kimball & Braun (2015)*; $n$ = 6; mean: 171.84 $\mu m^3$; SD: 28.66) and 169% that of the avian average (non-Sphenisciformes; $n$ = 36; mean: 141.97 $\mu m^3$; SD: 27.82). The volume of penguin RBCs differs significantly from the one of

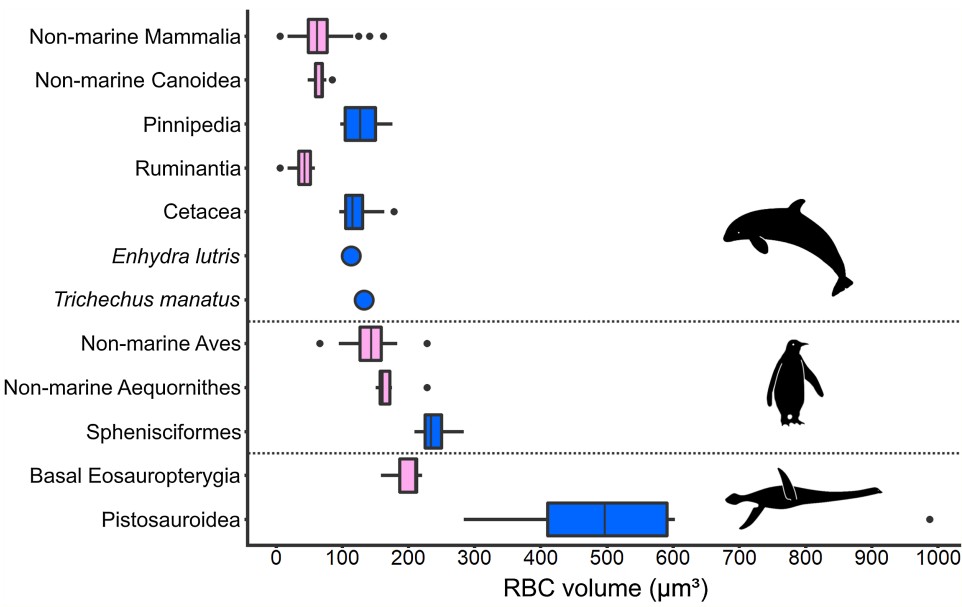

**Figure 5 Comparison of RBC size expressed as volume in amniotes displaying varying aquatic adaptation.** Above: Mammalia; Middle: Aves; Below: Sauropterygia. All three clades show an increase in RBC volume from terrestrial or shallow-water taxa (pink) to more aquatic, deep-diving taxa (blue). Silhouettes by Kai R. Caspar.

both other sea birds ($p = 0.002$) as well as from the avian mean ($p < 0.001$). In our sample, the only other bird within the penguin RBC volume range is the blue-eyed shag (*Phalacrocorax atriceps*), a deep-diving marine cormorant.

Data availability for extant marine reptile groups was far more restricted, consisting of data sets for either RBC area or RBC length and having incomplete taxonomic coverage. However, less extreme disparities between marine and non-marine reptile groups were observed compared to the endothermic amniotes. *Laticauda colubrina* is the only marine squamate within our dataset. With an RBC area of ca. 170 $\mu m^2$, it exhibits the largest RBCs of the family Elapidae (*Saint Girons, 1970*). *Laticauda* RBCs show 126% of the mean area reported for elapids ($n = 7$; range: 114.5–162 $\mu m^2$; mean: 134.5 $\mu m^2$) but are approached in area by those of terrestrial species such as *Pseudechis australis* (162 $\mu m^2$) (see Table S6). A similar pattern of limited size disparity depending on ecology was found for turtles. The mean lengths of marine turtle RBCs are significantly larger compared to those of freshwater cryptodire turtles (marine species: 22.74 $\mu m$, $n = 7$; non-marine species: 20.10 $\mu m$, $n = 20$, $p < 0.001$). Nevertheless, there is a size range overlap between the groups.

## DISCUSSION

### Methodological issues

Our analysis is strongly dependent on the availability of data for extant species. Since plesiosaurs display an extremely derived morphology and because the phylogenetic position of Sauropterygia has not been unequivocally determined, phylogenetic models for estimating trait values for this group may be biased. Our PEM model on RBC size parameters was based on RBC and vascular canal dimension data published by

*Huttenlocker & Farmer (2017)*. Although this study included representatives of all major amniote clades, overall diversity and species number in the sample is low. Our results are thus open to testing by expanding the given dataset for extant taxa. In addition, we suspect a phylogenetic influence on the generally larger size of RBCs in reptiles, to which plesiosaurs belong, compared to mammals and birds.

## RBC size evolution in eosauropterygians: effects of body mass, genome size and metabolic rate

RBCs show a notable difference in size (area as well as volume) between basal eosauropterygians and the pistosauroids. Species of pistosaurid grade have values intermediate between the basal groups and plesiosaurs. When compared to modern taxa, the inferred RBC parameters of eosauropterygians fall well within the range of extant non-avian reptiles which generally have the largest RBCs of all amniotes. While basal sauropterygians show inferred RBC sizes (area, length, and width) similar to the lowest values obtained for squamates, the inferred values for plesiosaurs indicate large RBCs, comparable in size to those of turtles or large lepidosaur RBCs (Table 1; Table S6). Our estimates therefore lie within a biologically reasonable range.

RBC size might be affected by body mass. Eosauropterygians cover a wide mass range, from the diminutive pachypleurosaurids to some plesiosaurs exceeding 10 m in length. Extant sauropsids demonstrate that RBC size and also cell sizes in various other tissues vary between individuals as well as between species of disparate mass (*Venzlaff, 1911*; *Hartman & Lessler, 1963*; *Frair, 1977*; *Kozłowski et al., 2010*; *Frýdlová et al., 2013*). For RBC area, a close correlation with body mass has been demonstrated for example in eublepharid geckos (*Starostová, Kratochvíl & Frynta, 2005*). Our own dataset on reptile RBCs also suggests that there is a weak but highly significant correlation across ectothermic amniotes in general and also in various lower ranking groups. Nevertheless, against the background of weak scaling effects, it appears that the observed patterns are too divergent to be exclusively related to body mass increase. Accordingly, we do not consider evolutionary body mass increase to be the major explanation for the difference in cell size parameters in basal versus derived eosauropterygians.

Several studies concluded that vertebrate genome size closely correlates with cell size (*Olmo & Odierna, 1982*; *Gregory, 2000*, *2001*), suggesting that an increase in genome size might have resulted in the enlargement of RBCs in the sauropterygian lineage. However, results of these studies have proven to be problematic, especially as cause and effect of the observed correlation remain obscure. Investigations on cell and genome size usually concentrate on high taxonomic ranks and often include only small samples from specific subgroups. General correlations between cell size and genome size might simply reflect physiological constraints acting in conjunction with non-adaptive fluctuations in genome size, without universal implications for specific taxa (*Pagel & Johnstone, 1992*; *Starostová, Kratochvíl & Flajšhans, 2008*). For example, a detailed study on RBCs and genome sizes in eublepharid geckos did not reveal a significant correlation between the two parameters (*Starostová, Kratochvíl & Flajšhans, 2008*). Similarly, inconsistent patterns are also known from other tetrapod groups, such as artiodactyls

(*Gregory, 2000*). Comparative data on cell and genome size at lower taxonomic ranks could potentially provide compelling evidence for a close correlation between the two or might elucidate the proposed link between genome size and cell size. The high mass-specific basal metabolic rate in pistosauroids (*Bernard et al., 2010*; *Krahl, Klein & Sander, 2013*; *Fleischle, Wintrich & Sander, 2018*), which is expected to correlate with a decrease in genome size (*Gregory, 2002*; *Kozlowski, Konarzewski & Gawelczyk, 2003*; *Vinogradov & Anatskaya, 2006*), sheds further doubt on the hypothesis that major genome expansions occurred during eosauropterygian evolution. As a consequence, we do not consider genome expansion a convincing explanation for evolutionary RBC size increase in Sauropterygia.

## Adaptive significance of secondarily enlarged RBCs in plesiosaurs and other marine amniotes

An increase in RBC size appears to be a ubiquitous, albeit not generally acknowledged, adaptation among secondarily aquatic amniotes. By comparing RBC parameters of marine groups with those of their respective non-marine relatives, we consistently found enlarged RBCs in the former, most prominently in mammals. In both birds and mammals, the largest RBCs incorporated in our dataset derive from pelagic specialists. The relative size increase of RBCs was largest in cetaceans which displayed on average 297% of the mean RBC volume found in the closely related ruminants. However, it should be noted that artiodactyls show an unusually broad spectrum of RBC sizes, including the smallest ones known in mammals (*Gregory, 2000*). This likely biased the relative RBC size increase recovered for cetaceans. Interestingly, our finding of consistently enlarged RBCs in marine amniotes calls a recent hypothesis on the hematology of ichthyosaurs into question. *Plet et al. (2017)* recovered microscopic disc-shaped structures from a Jurassic ichthyosaur vertebra encapsulated in a carbonate concretion and interpreted them as miniaturized RBCs. Like plesiosaurs, ichthyosaurs were large pelagic endotherms (*Bernard et al., 2010*), so that an increase rather than a reduction of RBC size in this taxon would be expected, based on our dataset.

Increasing RBC size might at first appear to be maladaptive in sustainably active aquatic endothermic animals. Enlarged RBCs are less effective than smaller cells in providing surrounding tissues with oxygen because of their reduced relative surface area, which restricts the diffusion of gas molecules (*Lay & Baldwin, 1999*; *Nicol, Melrose & Stahel, 1988*). With increasing RBC volume, the rate of oxygen uptake and release, respectively, within a specific time period is steadily reduced (*Holland & Forster, 1966*). However, since greater quantities of hemoglobin can be stored in each individual cell, larger RBCs can maintain tissue oxygen supply for a longer time interval than smaller ones at a constant hematocrit level (*Wickham et al., 1989*; *Promislow, 1991*). This is especially relevant for prolonged aerobic dives, facilitating foraging in pelagic habitats. Possibly, this advantage outweighs potential risks related to the formation of RBC aggregations, which are apparently tolerated by marine mammals to a degree hypercritical to terrestrial species (*Castellini et al., 2006*). The prevalence of enlarged RBCs in deep diving flying species, such as the blue-eyed shag, which would otherwise benefit from smaller cells, further

supports an adaptive value of this trait. Apart from this, enlarged RBCs have been controversially hypothesized to be advantageous for pelagic specialists by altering blood rheology (*Block & Murrish, 1974*; *Castellini et al., 2010*). In amniotes, RBC size is inversely correlated with RBC counts (*Hartman & Lessler, 1963*; *Hawkey et al., 1991*). Marine tetrapods exhibiting large RBCs therefore have low RBC counts compared to terrestrial taxa while exhibiting higher hematocrit values (*Nicol, Melrose & Stahel, 1988*; *Wickham et al., 1989*; *Hedrick & Duffield, 1991*). This condition was reported to reduce blood viscosity at relevant shear rates and proposed to aid in sustaining tissue perfusion and effective circulation during diving cycles (*Wickham et al., 1989*; *Clarke & Nicol, 1993*). However, other reports offer contrasting results (*Block & Murrish, 1974*; *Hedrick & Duffield, 1991*). Thus, currently available data fail to produce a conclusive picture of the matter (*Castellini et al., 2010*).

Interestingly, relative RBC size increase in extant marine reptiles appears to be far less pronounced than in endotherms. However, this conclusion is tentative and calls for further investigation, as informative data are extremely scarce. The limited data suggest that, as in endotherms, marine specialists among reptiles tend to evolve larger RBCs, but the size increase is far more limited. In turtles, a group in which individual as well as species-specific body mass has a notable influence on RBC size (*Frair, 1977*), the consistently larger body mass of marine species might additionally contribute to the observed cell enlargement compared to limnic groups. The potential difference in relative RBC size increase in endotherms and ectotherms could be linked to the divergent oxygen demands in the respective groups. However, this preliminary hypothesis requires support from further hematological studies on marine reptiles. Extant marine reptiles appear to be of limited use in the comparison with plesiosaurs, in particular, because of the differences in basal metabolic rate.

The largest RBCs in each group studied are predominately found in species that routinely dive to great depths such as sea elephants and bottom-feeding monodontid whales (*MacNeill, 1975*; *Hedrick & Duffield, 1991*). Following this pattern, phocid seals, which tend to dive deeper and for longer durations than their otariid relatives (*Debey & Pyenson, 2013*), have consistently larger RBCs. However, shallow-water inhabitants such as the Chinese river dolphin (*Lipotes vexillifer*) can exhibit remarkably large RBCs as well, while comparatively small cells can occur in deep diving species. For example, the king penguin (*Aptenodytes patagonicus*), which is among the most extreme avian divers, reaching depths of more than 300 m (*Kooyman et al., 1992*), displays the smallest RBCs within our penguin sample. Accordingly, there appears to be no tight correlation between diving depth and RBC size within a specific group. Further research needs to elucidate factors influencing RBC size variations within taxa of shared ecology. However, it can be robustly stated that relative taxon-wide RBC enlargement is associated with aquatic adaptation, at least in endotherms.

Given the collective evidence from extant species, we suggest that the demands posed by foraging in offshore environments and elevated metabolic rates drove the evolution of significantly enlarged RBCs in the Pistosauroidea. Following that, we hypothesize low RBC

counts and high hematocrit values in plesiosaurs, mirroring the situation found in penguins, whales, and seals.

The inferred RBC volume difference (270%) between basal eosauropterygians and pistosauroids is comparable to the difference between fully aquatic and terrestrial species seen in modern mammals. It is therefore more extreme than expected, given that pachypleurosaurids and especially Nothosaurus were already well adapted inhabitants of coastal waters. However, adaptive shifts in RBC size of, for example in *Nothosaurus*, were probably relatively small when compared with its terrestrial forerunners, as the size increase in modern reptilian analogs suggest. Subsequently, RBC size in more derived eosauropterygians of the pistosaurid clade notably increases, as did basal metabolic rate (*Fleischle, Wintrich & Sander, 2018*). As already described above, RBC size usually scales negatively with metabolic rate. In the specific case of endothermic aquatic amniotes, however, elevated respiratory demands apparently override the metabolic pressures limiting RBC size and favor enlarged cells. The optimized hematology of pistosauroids was gradually acquired in species of the pistosaurid grade. Therefore, its evolution coincided not only with the emergence of elevated metabolic rates but also changes in sensory ecology, locomotion, and diving profiles compared to pachypleurosaurids and nothosaurids (*Sues, 1987*; *Neenan et al., 2017*; *Surmik et al., 2017*). All of these traits prepared the emergence of the Plesiosauria in the Late Triassic. The divergent hematology of the groups concerned should therefore be viewed as another expression of ecological separation between them.

For species-specific RBC size interpretations, we suggest a cautious approach with respect to ecology. As noted above, RBC size variation does not correlate tightly with suggestive behavioral differences such as diving depth and duration in extant aquatic amniotes. This complicates detailed inferences for fossil taxa. For example, we estimated a remarkably large RBC size for the plesiosaur *Pliosaurus* sp. (Table 2; Fig. 3). While this could be interpreted as an indication of deep and prolonged diving in this genus, the inconsistent patterns observed in extant marine amniotes ask for more cautious considerations. A broader sampling of extant as well as fossil taxa is needed to convincingly evaluate ecological signals at the level of the genus or species.

## CONCLUSIONS

Our results support previous studies proposing an ecophysiological separation between basal eosauropterygians of coastal environments and the increasingly pelagic pistosauroids. Living in offshore habitats necessitates proficient diving abilities, which in consequence must have required physiological adaptations to prolonged submersion in pistosauroids. Large RBCs and the thereby enhanced constant oxygen supply to somatic tissues would have facilitated deep diving, which also is the case in numerous modern clades of pelagic amniotes. Estimates of RBC size in pistosauroids suggests remarkably large RBCs in this group, thereby supporting the view of especially plesiosaurs as predominately pelagic animals. The RBC size increase evolved simultaneously with the plesiosaurian bauplan in

basal pistosauroids and coincided with the emergence of a unique bone microstructure (*Krahl, Klein & Sander, 2013*) and compact inner ear morphologies (*Neenan et al., 2017*). All of these findings support the assumption that basal pistosauroids, such as *Cymatosaurus* and *Pistosaurus*, gradually adapted to an offshore lifestyle during the Middle Triassic. However, RBC size apparently does not represent a reliable proxy to infer specific ecological niches and diving depths, as comparisons with extant species demonstrate.

We suggest that studies on the hematology of other fossil groups have the potential to unveil and date the emergence of specific ecological adaptations and to test hypotheses put forward herein. We also encourage further studies on extant marine amniotes to allow for refined inferences for fossil taxa. So far, RBC evolution appears to represent a remarkable example of adaptive convergence between Mesozoic marine reptiles, oceanic mammals and pelagic diving birds.

## ACKNOWLEDGEMENTS

We thank Olaf Dülfer and Pia Schucht for producing the histological thin sections used in this study and all curators at the respective institutions for the permission for histological sampling. We are grateful to Shoji Hayashi and Yasuhisa Nakajima for providing the elasmosaurid the pliosaurid and elasmosaurid samples and for discussion. We are also very grateful to Lucas Legendre and Jorge Cubo for assistance with statistical modeling. We thank Sabine Begall and Jun Liu for critical feedback and comments during preparation of the initial draft and Bruce Rothschild as well as one anonymous reviewer for constructive criticism that improved the original manuscript.

### Funding

Funding was provided by the German Research Foundation (grant no. SA 469/47-1). The funders had no role in study design, data collection and analysis, decision to publish, or preparation of the manuscript.

### Grant Disclosures

The following grant information was disclosed by the authors:
German Research Foundation: SA 469/47-1.

### Competing Interests

The authors declare that they have no competing interests.

### Author Contributions

- Corinna V. Fleischle conceived and designed the experiments, performed the experiments, analyzed the data, prepared figures and/or tables, authored or reviewed drafts of the paper, approved the final draft, performed statistical modelling.
- P. Martin Sander performed the experiments, contributed reagents/materials/analysis tools, authored or reviewed drafts of the paper, approved the final draft.

- Tanja Wintrich performed the experiments, contributed reagents/materials/analysis tools, approved the final draft, revised the manuscript.
- Kai R. Caspar conceived and designed the experiments, performed the experiments, analyzed the data, prepared figures and/or tables, authored or reviewed drafts of the paper, approved the final draft, compiled biometric datasets on extant taxa.

## Data Availability

Original measurements and statistical notes as well as published biometric data analyzed are available in the Supplemental Files.

All petrographic thin-sections used in the study are housed at the Section Paleontology, Institute of Geosciences, Universität Bonn, Bonn, Germany, as part of the research collection of the Sander lab.

Accession numbers are as follows (one specimen per species was used):

*Anarosaurus heterodontus*: NMNHL Wijk. 06-38fe

*Neusticosaurus edwardsii*: PIMUZ T3455

*Neusticosaurus peyeri*: PIMUZ T 4089

*Neusticosaurus pusillus*: PIMUZ T 3566

*Nothosaurus* sp.: IGWH 21

*Cymatosaurus* sp.: IGWH 6

*Pistosaurus longaevus*: SMNS 84825

*Cryptoclidus eurymerus*: IGPB R 324

*Elasmosauridae indet.*: OMNH MV 85

*Plesiosaurus dolichodeirus*: IGPB R90

*Pliosaurus* sp.: SMNS 96896

*Polycotylus latipinnus*: LACM 129639A

*Rhaeticosaurus mertensi*: LWL MfN P 64047

## Supplemental Information

Supplemental information for this article can be found online at http://dx.doi.org/10.7717/peerj.8022#supplemental-information.

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
