# Peer review of "Hematological convergence between Mesozoic marine reptiles (Sauropterygia) and extant aquatic amniotes elucidates diving adaptations in plesiosaurs"

_PeerJ, doi:10.7717/peerj.8022_

## Round 0.1 · original submission · Minor Revisions

Dear authors,

I have accepted the decision of 'minor revisions' from both of the reviewers. The primary comments relate to the incorporation of data from Rothschild and Storrs (2003), and the interpretations for pistosauroids.

I look forward to receiving your revised manuscript.

·

Basic reporting

The article is well written, unambiguous and uses professional English throughout. Introduction and background clearly describes the Sauropterygia and the issues involved in their bauplan, lifestyle and habitat utilization, utilizing multiple support including inner ear morphology and RBC size derived from various measures of bone capillary circulation. Structure conforms to PeerJ standards and figures are relevant, high quality, and well labeled and described. Supplement provides appropriate raw data.

This further documents and clarifies the adaptation of this group to diving behavior. Underlying data is robust and sound.

Experimental design

Experimental design is original primary research within scope of the journal. The research question is well defined, relevant and meaningful and identifies how it fills an identified knowledge gap. It has been performed to a high technical and ethical standard, including discussion and citation of articles with alterative explanations or outliers.

Validity of the findings

Conclusions are well stated, linked to the original research question and limited to supporting results.

Additional comments

I commend the authors for their multi-disciplinary approach. The manuscript is clearly written in professional, unambiguous language. I find no weakness, but would suggest one minor alteration to figure 3. Perhaps consider incorporation of the data (Rothschild and Storrs 2003) that they cited:
Perhaps enter the prevalence information of avascular necrosis evidence for deep diving”
Neusticosaurus 4/346
Nothosaurus – 0/2000
Pliosaurus – 3/7
Plesiosaurus 7/30
Cryptoclidus 1/23
Polycotylus 0/9
Elasmosauridae 3/7

Reviewer 2 ·

Basic reporting

no comment

Experimental design

no comment

Validity of the findings

no comment

Additional comments

This is a very exciting and intriguing study. I found the discussion to be balanced and engaging and well written. A minor but important point that should be included in the discussion is that smaller RBC's are tightly correlated with higher aerobic athleticism in vertebrates that use either terrestrial locomotion or that fly. Larger RBC's in marine groups are generally thought to be the result of a reduced cost of transport in swimming animals compared to flying or running vertebrates. Thus, the finding of larger RBC's in pistosauroids is mysterious. The explanation that more hemoglobin is stored in larger cells is not compelling, or needs to be better justified in the discussion, for the following reason. It would seem that it is the total hemoglobin in the blood, which is a function of the hematocrit, and not simply the hemoglobin content of each cell, that would be important for the blood's capacity for oxygen storage. Thus many small RBC's would be as effective for oxygen storages as fewer, larger RBCs if hematocrit is constant. Furthermore, as mentioned in the manuscript, large RBC's will slow gas exchange due to the reduced surface area to volume ratio. Thus, it would seem, there must be another explanation for the enlarged cells.

I don't claim to have the answer, but it would be nice to hear some alternative explanations. Could the enlarged RBC's be related to the time spent at the surface replenishing oxygen stores? If pistosauroids spent more time breathing at the surface than other groups, there might be relaxed selection for rapid transport of gas between blood and air. Could it be that they were selected more for locomotor economy than sustained high speed swimming? Or perhaps greater efficiency of locomotion reduced the energetic cost of swimming? At any rate, the idea of greater oxygen carrying capacity of the blood being the driver for enlarged RBC's is not very convincing, unless, as was stated, this is related to changes in viscosity. However, the relationship with viscosity was not well developed, and could use more attention.

Another minor point is that it would be nice to see snakes separated by a different symbol figure 4, particularly since they have the opposite slope.

---

## Round 0.2 · Minor Revisions

Dear authors,

I have accepted the decision of ‘minor revisions’. As you will see there is only one small grammatical issue to address.

Reviewer 2 ·

Basic reporting

The paper would be improved if on line 104 the authors replace 'detailly studied' with 'studied in detail'. Detailly is not a word. Otherwise, the paper is ready for publication.

Experimental design

no comment

Validity of the findings

no comment

Additional comments

No comment

---

## Round 0.3 · accepted · Accept

Dear authors,

Many thanks on your return of the revised manuscript. I can confirm that it has been accepted for publication.

You will shortly be contacted by PeerJ production staff about the proofs.

Thank you again for choosing PeerJ as your publication venue, and we hope you use us again in the future.